# Primary Amenorrhea with Apparently Absent Uterus: A Report of Three Cases

**DOI:** 10.3390/jcm11154305

**Published:** 2022-07-25

**Authors:** Eva Porsius, Marian Spath, Kirsten Kluivers, Willemijn Klein, Hedi Claahsen-van der Grinten

**Affiliations:** 1Department of Pediatric Endocrine Disease, Amalia Children’s Hospital, Radboud University Medical Centre, P.O. Box 9101, 6500 HB Nijmegen, The Netherlands; hedi.claahsen@radboudumc.nl; 2Department of Obstetrics and Gynaecology, Radboud University Medical Centre, P.O. Box 9101, 6500 HB Nijmegen, The Netherlands; marian.spath@radboudumc.nl (M.S.); kirsten.kluivers@radboudumc.nl (K.K.); 3Department of Medical Imaging, Radiology, Radboud University Medical Centre, P.O. Box 9101, 6500 HB Nijmegen, The Netherlands; willemijn.klein@radboudumc.nl

**Keywords:** female genital malformations, primary amenorrhea, absent uterus, Müllerian agenesis, DSD

## Abstract

Background: The apparent absence of a uterus upon imaging women with primary amenorrhea appears to lead to a high risk of misdiagnosis, which will lead to significant mental distress in patients. Case: Three young females with primary amenorrhea were referred with a diagnosis of Mayer–Rokitansky–Kuster–Hauser syndrome based on radiological findings of an apparently absent uterus. In two patients, the absence of the uterus could be confirmed, but with various diagnoses. The other patient had a normal but unstimulated uterus due to her hypoestrogenic state. Summary and Conclusion: The presented cases illustrate the broad differential diagnoses and the specific pitfalls of primary amenorrhea with an apparently absent uterus upon imaging. A well-established diagnosis was only possible through a thorough correlation of imaging findings with clinical history, biochemical findings and physical examination.

## 1. Introduction

Primary amenorrhea is a common problem that leads to severe distress in adolescents. Normal puberty starts with breast development at the age of eight to thirteen years. Menarche usually occurs two years after the first signs of puberty, at age eleven to fifteen years. Primary amenorrhea is defined as the absence of menarche by age fifteen or more than three years after the start of breast development [1]. The cause of primary amenorrhea can be due to anatomical abnormalities of the uterus or its outflow tract, and endocrine disorders of the gonads, pituitary gland or hypothalamus [2,3]. The presence or absence of a uterus can help to categorize primary amenorrhea, as illustrated in Figure 1.

Uterus absence in otherwise healthy females with normal external genitalia can be caused by different conditions within differences in sex development (DSD) conditions. Therefore, a DSD condition, such as Müllerian agenesis, testosterone synthesis defect or complete androgen insensitivity syndrome (CAIS), should be considered as the underlying cause of primary amenorrhea.

Pelvic ultrasound and magnetic resonance imaging (MRI) are good methods to visualize Müllerian structures [3,4]. When the uterus cannot be detected by ultrasound, MRI is advised for detailed imaging of Müllerian structures, ovaries, the vagina and kidneys [3]. It has to be considered that, especially in prepubertal state, a present uterus may remain undetected, since in the hypoestrogenic state the uterus is small and not easy to visualize. The interpretation of imaging findings by an experienced radiologist with knowledge of the normal appearance of the uterus and ovaries during different phases of growth and development is therefore preferred [4].

In this paper, we present three cases of young women who were referred to our hospital presenting with primary amenorrhea and an apparently absent uterus. Our aim is to point out the importance of correlating imaging findings with clinical history, biochemical findings and physical examination, and provide guidance for distinguishing the various diagnoses of primary amenorrhea with an absent uterus upon imaging. This is important because these cases are rare, but various medical practitioners could encounter them and misdiagnosis will lead to significant mental distress in patients.

## 2. Case 1

A seventeen-year-old girl was referred to the local gynecologist because of primary amenorrhea. Puberty had started with breast development and pubarche at the age of ten. She had used the oral contraceptive pill, but no withdrawal bleeding ever occurred. According to the patient, penetration during sexual intercourse was possible without pain. However, from a physical examination and vaginal ultrasound, a short and blind-ending vagina was found. The MRI suggested an absent uterus, normally located ovaries and normal kidneys (Figure 2). With the suspicion of Mayer–Rokitansky–Küster–Hauser syndrome (MRKHS), the patient was referred to our expert center. An MRI demonstrated no vaginal, myometrial or cervical tissue, with some fibrous and lipomatous tissue at this location. The ovaries had normal cystic appearance and were situated in the broad ligament. Karyotype was 46,XX. A hormonal examination showed normal endocrine parameters (Table 1). With these results, a diagnosis of MRKHS was confirmed and the patient received information about the etiology and consequences of this diagnosis.

## 3. Case 2

A seventeen-year-old girl presented with primary amenorrhea after a normal start to puberty. Breast development started at the age of twelve years and she had a growth spurt two years later. The physical examination showed the normal anatomy of the vulva and shaved pubic hair, and the vaginal depth upon digital palpation was 3–4 cm (reference 7–15 cm [5]). The uterus and ovaries could not be visualized by pelvic ultrasound (Figure 3a). MRI performed in the local hospital confirmed the absence of a normal uterus and presence of a short vagina. In this MRI report, normal ovaries were described. These results lead to the assumption of an MRKHS diagnosis. This was explained to the patient and vaginal dilation therapy was suggested.

She was referred to our expert center with suspicion of MRKHS. In the physical exam, we noted her tall stature (height +1.17 SDS) and very sparse axillary and pubic hair. After the specialized radiologist reviewed the MRI, the previously described ovaries actually appeared as atypical gonadal structures with a large solid part, suspected of being (ovo)testicular tissue (Figure 3b). A hormonal examination showed high levels of anti-Müllerian hormone (AMH) and testosterone (Table 1). Genetic diagnostics confirmed the suspicion of a 46,XY karyotype and an androgen receptor gene mutation (hemizygous variant NM_000044.2(AR):c.2594A>T, p.(Asp865Val)), confirming the diagnosis of CAIS. Follow up by our DSD team is continued to guide her during vaginal dilation therapy, offer psychological support and screening for a possible malignant transformation of the abdominal testes.

## 4. Case 3

A sixteen-year-old girl presented with delayed puberty and primary amenorrhea. Breast development started at age fourteen, but no further progression occurred after tanner stages 1–2. Axillary and pubic hair also developed up to tanner stage 2. There were no signs of virilization. She had a short stature and low weight (Table 1). Bone age was delayed 3 years according to Gruelich and Pyle using the automatic BoneXpert method [6]. There were no signs of eating disorders, malabsorption or other chronic illness. The mother had normal puberty with menarche at twelve years old. The father experienced late but spontaneous puberty. There was no history of medication use, except sporadic corticosteroid ointment for eczema. Hormonal serum analysis showed low FSH, LH and estradiol levels, normal thyroid function and prolactin levels (Table 1). The karyotype was 46,XX. Abdominal ultrasound performed at the local hospital showed no uterus or ovaries. Subsequent MRI showed normal ovaries and a small uterus (2.3 × 1.3 cm), described as a rudimentary uterus suspected for MRKHS. 

She was referred to our expert center with a diagnosis of MRKHS. Ultrasound and MRI revision by our radiologist showed a small uterus with visible myo- and endometrial tissue, which was interpreted as an unstimulated hypoestrogenic uterus (Figure 4). A further diagnostic evaluation of the etiology of hypogonadotropic hypogonadism was performed. Cerebral MRI showed no pathology in the hypothalamus or pituitary gland. Exome sequencing known genes connected to hypogonadotropic hypogonadism showed no mutations. Pubertal induction with estrogen therapy was started, which resulted in breast development, endometrial thickening and menarche.

## 5. Discussion

This study presents three patients with primary amenorrhea, who were referred to our expert clinic with a (mis)diagnosis of MRKHS because of radiological findings of an apparently absent uterus. In two patients, the absence of the uterus could be confirmed, but with various diagnoses (MRKHS resp. CAIS). The other patient had a normal but unstimulated uterus. Our case report shows that the apparent absence of a uterus upon imaging in women with primary amenorrhea appears to lead to a high risk of misdiagnosis. The presented cases illustrate the broad range of diagnoses of primary amenorrhea and specific pitfalls. A well-established diagnosis was only possible through considering the combination of clinical history, physical examination, biochemical evaluation and assessment of imaging by an experienced radiologist.

During the prenatal development of 46,XX individuals, the Müllerian ducts fuse in absence of AMH to form the uterovaginal canal. In syndromes with Müllerian agenesis, such as MRKHS, this process fails, resulting in the absence of the uterus and upper part of the vagina [7]. In 46,XY individuals, AMH stimulates the regression of Müllerian ducts, resulting in absence of the uterus. Testosterone and dihydrotestosterone are responsible for the virilization of external genitalia. The effect of these androgens is compromised in androgen receptor defects and testosterone synthesis or metabolizing defects. A 46,XY female with CAIS but normal functioning AMH will thereby have external female features in the absence of a uterus and will have testes instead of ovaries [8].

The first step in the evaluation of primary amenorrhea is carefully looking through clinical history and physical examination. Women with MRKHS experience a normal start of puberty with age-appropriate breast development and pubarche, but primary amenorrhea as an isolated symptom [7]. In CAIS, there is a normal onset of breast development, but no or very sparse axillary and pubic hair. A Tanner stage examination is important since patients will often unknowingly misinterpret pubarche. Height is taller than general. Acne and sweat odor are absent. Breasts and female adiposity can develop through the action of estradiol deriving from the peripheral aromatization of testosterone [8]. Delayed pubertal development in 46,XX individuals with an absence of breast development and menarche is caused by estrogen absence, due to ovarial or pituitary/hypothalamic causes. 

Thus, assessing patient history and physical findings carefully an accurate diagnosis can be made. However, as the impact of these diagnoses is great, biochemical analysis, imaging or karyotype analysis is conducted to support this. A careful consideration of necessary diagnostics will prevent misinterpretation and misdiagnosis.

LH, FSH, progesterone, estradiol, TSH and prolactin can guide in finding possible endocrine causes of primary amenorrhea [2]. In response to our cases, we suggest adding AMH and testosterone to give direction in determining possible 46,XY DSD conditions. Karyotype analysis should always be performed when 46,XY DSD conditions are considered.

In women with normal secondary sexual characteristics but primary amenorrhea, pelvic ultrasound is advised to search for possible anatomical causes [2,3]. In case of an apparently absent uterus on ultrasound, we advise performing MRI and consulting a specialized radiologist. In CAIS patients, no Müllerian structures will be found and gonadal tissue might be testicular or maldifferentiated. In women with an absent uterus because of MRKHS, absence might not always be a full absence. Wang et al. [9] described bilateral Müllerian rudiments seen on MRI in 95% of MRKHS women and normally located ovaries in 68.7%. The Müllerian rudiments are generally small with only one-layer differentiation. There is also a high incidence of renal (~30%) and skeletal (~10–40%) anomalies [7].

In women with delayed breast development, uterus imaging is not advised, since mostly estrogen deficiency will be the cause of primary amenorrhea. An important consideration is that, in the hypoestrogenic state, the uterus is small and unstimulated and can therefore be difficult to visualize. Berglund et al. [10] described the association of an apparently absent uterus with primary ovarian insufficiency in their review of the literature. Estrogen deficiency has previously resulted in misdiagnosis as MRKHS in 22/25 patients (88%) with primary ovarian insufficiency. Michala et al. [11] demonstrates this as well in their description of ten patients with misdiagnoses of uterine agenesis based on imaging and laparoscopic findings. They suggest that imaging should be undertaken by clinicians with experience in management of this age group and in some girls it may be necessary to delay final diagnosis until after puberty or the administration of estrogen therapy.

In case of absent uterus upon imaging, a progestin challenge test [3] will probably not lead to vaginal bleeding. In modern practice, there seems to be no additional value for this test, but a high risk of further delay of correct diagnosis. 

The diagnosis of an absent uterus and vagina leads to a negative impact on the affected women’s level of psychological distress and self-esteem [12]. Additionally, these women are confronted with sexuality and fertility issues at a young age. In the case of the misdiagnosis of MRKHS instead of CAIS, the adolescent will be confronted with confusing information: first the absence of uterus and vagina and subsequent information on afunctional gonads. This has even further fertility consequences and, therefore, it is important to make an early correct diagnosis. Confirmation by a DSD team is advised [3]. This specialized team consists of a gynecologist, endocrinologist, radiologist, psychologist, geneticist, urologist and sexologist.

## 6. Conclusions

The presented cases illustrate the broad differential diagnosis and the specific pitfalls of primary amenorrhea with an apparently absent uterus upon imaging. It is important not to draw premature conclusions from findings of an apparently absent uterus from imaging alone. A well-established diagnosis was only possible with a thorough correlation of imaging findings with clinical history, biochemical findings and physical examination.

## Figures and Tables

**Figure 1 jcm-11-04305-f001:**
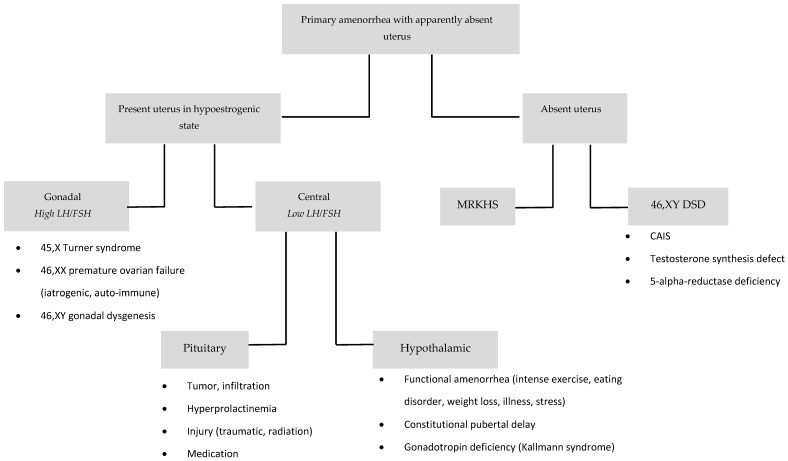
Differential diagnoses of primary amenorrhea with apparently absent uterus on imaging in women with normal external genitalia. LH: luteinizing hormone; FSH: follicle-stimulating hormone; MRKHS: Mayer–Rokitansky–Küster–Hauser syndrome; DSD: differences in sex development; CAIS: complete androgen insensitivity syndrome.

**Figure 2 jcm-11-04305-f002:**
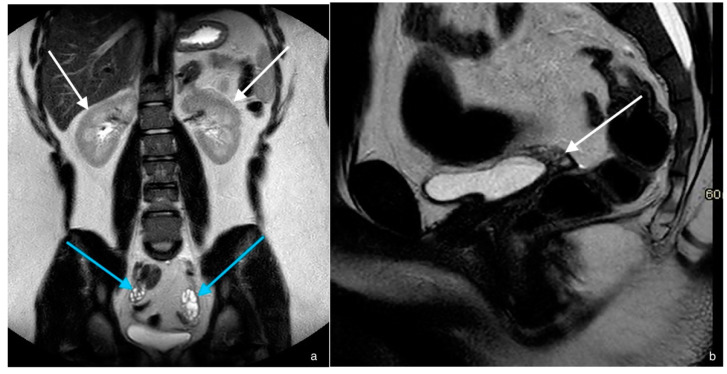
Case 1. (**a**) Coronal T2-weighted image of the abdomen, showing two normal kidneys (white arrows) as well as two normal ovaries (blue arrows). (**b**) Mid-sagittal T2-weighted image of the pelvis, demonstrating a triangular shape between bladder and rectum, which is the fibro-adipose remnant tissue of the aplastic uterus, without any endometrial or myometrial organization (white arrow).

**Figure 3 jcm-11-04305-f003:**
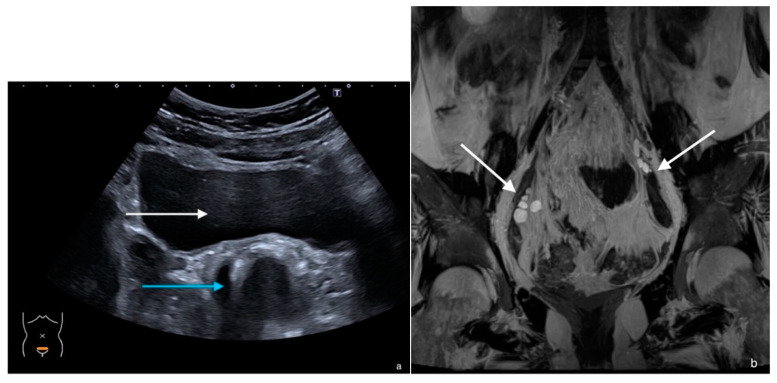
Case 2. (**a**) Ultrasound of the pelvis, demonstrating the rectum (blue arrow) direct dorsal to the bladder (white arrow), without uterus in between. (**b**) Coronal T2-weighted MR image of the pelvis, with the white arrows indicating the gonads with partially solid tissue and partially cystic, a suspected aspect of ovotestes.

**Figure 4 jcm-11-04305-f004:**
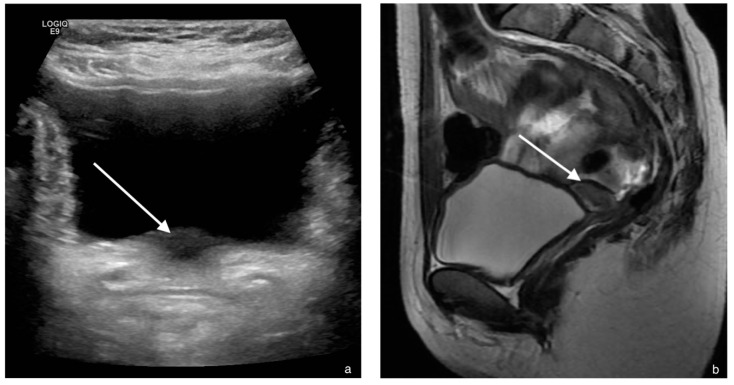
Case 3. (**a**) Ultrasound of the pelvis. The white arrow indicates a small uterus behind the bladder. (**b**). Mid-sagittal T2-weighted MR image, demonstrating a small uterus (white arrow). (**c**) The oblique-coronal T2-weighted images demonstrate clearly that there is an architecture of endometrial and myometrial tissue (white arrow).

**Table 1 jcm-11-04305-t001:** Overview of important findings in three cases with primary amenorrhea, based on Mayer–Rokitansky–Küster–Hauser syndrome (MRKHS), complete androgen insensitivity syndrome (CAIS) and hypogonadotropic hypogonadism of unknown origin.

		Case 1	Case 2	Case 3
Diagnosis		MRKHS	CAIS	Hypogonadotropic hypogonadism e.c.i.
History		Normal start of breast development and pubarche	Normal start of breast development, no pubarche	Delayed and incomplete breast development, normal pubarche
Physical examination		Tanner M4 P4 Female external genitalia with no virilization Short blind-ending vagina	Tanner M5 P1 Female external genitalia with no virilization. Tall stature. No axillary hair	Tanner M1-2 P2-3 Female external genitalia with no virilization BMI 15.8 kg/m^2^ (−2.2 SD) Height 1.56 m (−1.78 SD)
Biochemical analysis	reference values			
LH	2.4–12.6 E/L (follicular), 14.0–95.6 (ovular), 1.0–11.4 (luteal)	38.8	24	0.5
FSH	3.5–12.5 E/L (follicular); 4.7–21.5 (ovular); 1.7–7.7 (luteal)	6.6	2.8	1.2
Estradiol	45–854 pmol/L (follicular); 151–1461 (ovular); 82–1251 (luteal)	470	78	<18
AMH	0.52–12.01 ug/L	2.5	1292	10.1
Testosterone	0.52–2.0 nmol/L		13	
Progesterone	<1.3 (follicular); 1.3–12 (mid cycle); 19–120 (luteal) nmol/L	1.7	<0.25	
Prolactin	100–760 mE/L			250
TSH	0.27–4.20 mE/L			2.5
Karyotype		46,XX	46,XY	46,XX
Imaging findings	ultrasound	No visualization of uterus or ovaries	No visualization of uterus or ovaries	No visualization of uterus or ovaries
	MRI report on primary investigation	Absent uterus, impression of hypoplastic vagina. Normal multicystic ovaries	Absent uterus and presence of a short vagina. Normal ovaries	Normal multicystic ovaries and a small uterus of 2.3 × 1.3 cm, described as rudimentary uterus
	MRI after referral	Lipofibromatous tissue at location of the uterus, no endo- myometrial or cervical structures. Normal multicystic ovaries	Underdeveloped Müllerian structures. Gonads with (ovo)testicular aspect	Small uterus with normal endo- and myometrial tissue, normal multicystic ovaries

BMI: body mass index; SD: standard deviation; MRI: magnetic resonance imaging; LH: luteinizing hormone; FSH: follicle-stimulating hormone; AMH: anti-Müllerian hormone; TSH: thyroid-stimulating hormone.

## Data Availability

Not applicable.

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
