# Peer review of "Primary Amenorrhea with Apparently Absent Uterus: A Report of Three Cases"

_jcm, 2022, doi:10.3390/jcm11154305_

Round 1

Reviewer 1 Report

This paper reports 3 cases of young women who were referred to the hospital presenting with primary amenorrhea and an apparently absent uterus. They were initially all suspected of MRKH, but 2 of them had other final diagnoses (CAIS and hypogonadotropic hypogonadism). The authors contended that it is important to correlate imaging findings with clinical history, biochemical findings and physical examination to avoid mental distress in patients caused by misdiagnosis. Differentiating sex-differentiated disease is sometimes difficult, and it is useful for many clinicians to summarize the steps involved.
However, I concern some issues as listed below for publication.

Major concerns:
1. It seems that the new point of this paper is not clear. The authors should clearly indicate what the novelty of this paper is.
2. If possible, key images (at least either ultrasound or MRI) of each case should be included in the paper, to enhance the readers' understanding.

Minor concerns:
1. Is there a mix of dots and commas in the decimal points used in the numbers listed in Table 1? Please correct the shaky notation.

Reviewer 2 Report

Page 1:

My suggestion is to use the definition of primary amenorrhea from ESHRE International evidence based guideline for the assessment and management of polycystic ovary syndrome 2018

Change: Complete androgen insensitivity to Complete androgen insensitivity syndrome (CAIS)

Page 2 and further pages: Change: MRKH to MRKHS (Mayer-Rokitansky-Kuster-Hauser syndrome)

Page 2 and further pages: Correct karyotypes: 45 X to 45,X,   46 XX to 46,XX, 46 XY to 46,XY

Page 3: Did the patient (case 2) have ovotesticular tissue? It means the diagnosis of ovotesticular DSD. What type of AR gene mutation was identified in the patient?

Page 3: Case 3: The results of LH-RH stimulation test should be presented.

Page 5 lines 169-171: In my opinion progesterone level, in contrast to AMH or testosterone, is not useful in diagnosing possible endocrine causes  of primary amenorrhea. Why do you suggest to measure progesterone level?  

Page 6: In case of absent uterus on imaging, a progestin challenge test as well as  estrogen-progestin challenge test are useless.

Considering the subject the authors should mention Michala L, Aslam N, Conway G, Creighton S. The clandestine uterus: or how the uterus escapes detection prior to puberty. BJOG 2010;117:212–215.

Round 2

Reviewer 1 Report

The authors have appropriately addressed the points raised by the reviewer and revised the paper. The revisions were effective, and the paper is now clearer and more meaningful.